# Polymer/Silica Hybrid Waveguide Thermo-Optic VOA Covering O-Band

**DOI:** 10.3390/mi13040511

**Published:** 2022-03-25

**Authors:** Yuexin Yin, Mengke Yao, Yingzhi Ding, Xinru Xu, Yue Li, Yuanda Wu, Daming Zhang

**Affiliations:** 1State Key Laboratory of Integrated Optoelectronics, College of Electronic Science and Engineering, Jilin University, Changchun 130012, China; yxyin20@mails.jlu.edu.cn (Y.Y.); yaomk20@mails.jlu.edu.cn (M.Y.); dingyz20@mails.jlu.edu.cn (Y.D.); xuxr20@mails.jlu.edu.cn (X.X.); liyue19@mails.jlu.edu.cn (Y.L.); 2College of Materials Science and Opto-Electronic Technology, University of Chinese Academy of Sciences, Beijing 100049, China; wuyuanda@semi.ac.cn

**Keywords:** optical devices, multimode interference, Mach–Zehnder interferometer, variable optical attenuator

## Abstract

In this paper, a polymer/silica hybrid waveguide thermo-optic variable optical attenuator (VOA), covering the O-band, is demonstrated. The switch is fabricated by simple and low-cost direct ultraviolet (UV) lithography. The multimode interferences (MMIs) used in the Mach–Zehnder interferometer (MZI)-VOA are well optimized to realize low loss and large bandwidth. The VOA shows an extinction ratio (ER) of 18.64 dB at 1310 nm, with a power consumption of 8.72 mW. The attenuation is larger than 6.99 dB over the O-band. The rise and fall time of the VOA are 184 μs and 180 μs, respectively.

## 1. Introduction

To meet the ever-increasing demands of high-capacity communication, wavelength-division-multiplexing (WDM) techniques are widely researched and deployed for optical communications [1,2,3]. The total bandwidth is boosted, with the wavelength number (channels) increasing in the WDM system, via the arrayed waveguide grating (AWG) [4,5]. Meanwhile, the optical signal power level difference deteriorates the optical signal-to-noise ratio (OSNR) and bit error rate (BER). The variable optical attenuator (VOA) is a critical component and effective method to preserve the power uniformities across all signal channels. The integration of the AWG and VOA arrays not only achieves wavelength blocking and equalizing, but it also enhances the reliability and reduces the cost [6,7]. The power consumption of the VOA array is a challenge to their marriage on a single chip because of the temperature sensitivity of the AWG. Among all kinds of VOAs demonstrated [8,9,10,11,12,13,14,15], polymer-based planar lightwave circuit (PLC) is an ideal platform to realize large scale integration, due to a large transparent window, low power consumption, and easy fabrication [16,17,18,19,20,21,22,23,24,25,26,27,28]. These silicon-based optical devices offer low power consumption and fast speed, but technical issues, such as temperature dependence, polarization dependence, and optical fiber coupling, remain challenges for commercialization. Notably, the WDM system is operated under a large bandwidth. Especially coarse WDM (CWDM) in the O-band shows the center wavelengths with 1270 nm (lane 0), 1290 nm (lane 1), 1310 nm (lane 2), and 1330 nm (lane 3), respectively. VOA should show a large bandwidth cooperating with the WDM system. However, broadband polymer-based VOA is less discussed. In this paper, we design and fabricate a polymer/silica hybrid waveguide thermo-optic VOA, based on Mach–Zehnder interferometers (MZIs), covering the O-band. The multimode interferences (MMIs) used in the VOA are optimized to achieve low loss and large bandwidth. The VOA shows an extinction ratio (ER) of 18.64 dB at 1310 nm under 8.72 mW. An attenuation is larger than 6.99 dB covering the O-band. The rise and fall time of the VOA are 184 μs and 180 μs, respectively.

## 2. Design and Simulation

The polymer/silica hybrid waveguide consists of a silica buffer layer, a Poly (methy-methacrylate) (PMMA) cladding, and an SU-8 core, whose refractive indices are 1.4456, 1.4769, and 1.5812 at 1310 nm, respectively. With a high thermo-optic (TO) coefficient of 1.86 × 10^–4^ K^−1^, an SU-8 2002 (from Microchem Corporation) core enables low power consumption. Compared with all-polymer devices, silica buffer layer offers an immiscible buffer, reducing the solubility phenomenon problem of waveguide edges due to annealing. Besides, the large thermal conductivity (1.4 W K^−1^ m^−1^) of silica improves the speed response. The waveguide dimensions are chosen to be as small as possible while still being reliably producible with fabrication. A square waveguide of 3 × 3 μm^2^ is chosen. The effective refractive index is 1.5611 under the fundamental transverse electric (TE) mode. The VOA is simulated through a three-dimensional beam propagation method (3D-BPM). To achieve a large bandwidth, the 3 dB splitter used in MZI should split equally over a large wavelength region. Compared with directional couplers (DC) and Y-branch splitters, MMI offers large optical bandwidth, a compact footprint, and high fabrication tolerance. The loss caused by modal mismatch, between the MMI region and input/output waveguides, can be decreased by taper waveguides. The schematic of the optimized MMI is shown in Figure 1a. Width of the MMI *W_mmi_* is 20 μm. The gap between output waveguides *W*_3_ is 6 μm. According to self-image theory [29], the length of MMI *L_mmi_* is 250 μm. With both input and output ports, all three tapers are linear and widen from *W*_1_ = 3 μm to *W*_2_ = 4.5 μm, with the length of *L_taper_* = 6 μm. An SEM image of the fabricated MMIs with tapered ports is shown in Figure 1b. The field distribution of the optimized MMI is shown, in Figure 1c, to be under 1310 nm. The calculated spectra of the MMI are shown in Figure 1d. The transmissions of two output waveguides are equal, and they are −3.03 dB at 1310 nm. The excess loss of the MMI is 0.03 dB. The lowest transmission over the O-band is −3.64 dB.

The schematic of the MZI-based broadband VOA is shown in Figure 2a. Two MMI-based splitters (combiners) are connected through bend and straight waveguides. The gap between two modulation arms is 50 μm, which is enlarged from MMI’s output tapers *W*_3_ through two bend waveguides. The radius of two bend waveguides is 4000 μm in order to realize a low bend loss. A 2000-μm-length microheater is fabricated, above a single modulation arm, to tune the VOA. To improve the coupling efficiency, the widths of input and output waveguides are increased from 3 μm to 16 μm through the 200-μm-length linear tapers. We change the temperature of the modulation arm under the microheater to simulate the performance of the VOA. As shown in Figure 2b, the insertion loss (IL) is 0.03 dB, while Δ*T* = 0 K. The minimum transmission, −38.85 dB, happens while Δ*T* = 1.8 K. The inserts of Figure 2b show the field distributions under temperature changes of 0 and 1.8 K, respectively. Calculated spectra of the VOA, while the temperature change of the modulation arm is 0 K and 1.8 K, are shown in Figure 2c. While the temperature change is 0 K, the insertion loss is 0.03~1.66 dB over the O-band. While the temperature change is 1.8 K, the attenuation is larger than 18.76 dB. The maximum attenuation is 74.42 dB at 1302 nm.

## 3. Fabrication and Characterizing

To fabricate the proposed VOA, a 15-μm-thick SiO_2_ buffer layer was thermally oxidized on a Si substrate. Then, the negative photoresist SU-8 2002 was spin-coated as the core layer on the silica buffer layer, at a rotational speed of 600 r/min, and pre-baked with two steps, including 10 min baking at 60 °C and 20 min at 90 °C. Then, the coated chip was exposed to ultraviolet (UV) light from a mercury discharge lamp (peak emission wavelength, 365 nm; irradiance, 23 mW/cm^2^, ABM-USA, Inc., San Jose, CA, USA) for 4 s and post-baked with two steps, including 10 min baking at 65 °C and 20 min at 95 °C. Time of exposure should be adjusted for a good pattern-transfer, from the photomask to the SU-8 thin film, according to the energy density. After that, we removed the unexposed SU-8 polymer and hard-baked it at 120 °C. Finally, a 7-µm-thick PMMA upper cladding was spin-coated on the waveguide cores and baked at 125 °C for 2.5 h. A 500-nm-thick aluminum film was thermally evaporated over the upper-cladding. A 20-μm-width and 2000-μm-length microheater was formed, above the modulation arm, via another UV photolithography and wet etching process. The fabricated VOA is shown in Figure 3a. To characterize the fabricated VOA, the light from the tunable laser source (TSL-550, Santec, Japan) is launched into the device by the edge coupling system with single mode fiber (SMF). The polarization of light output from the tunable laser is adjusted by the polarization controller. The output power is monitored by an optical power meter (MPM200, Santec, Japan). The switch is tuned by applying electric power using source meter instruments (Keithley 2450, Tektronix, Beaverton, OR, USA). When applying the driving voltage on the microheater, the current and the transmission at 1310 nm are measured, as shown in Figure 3b,c. After a linear fitting of the current-voltage (I-V) curve, the microheater shows 1.745 Ω. An extinction ratio (ER) of 18.64 dB is measured while applying 8.72 mW power to the microheater. Then, we measure the normalized transmission spectra of the VOA, under 0.00 mW and 8.72 mW, as shown in Figure 3d. The fiber-to-fiber loss of 12.43 dB includes the propagation loss and coupling loss. The propagation loss mainly results from the material absorption in the near-infrared wavelengths region and surface scattering. The coupling loss primarily comes from the surface roughness of the end facet and the fiber-waveguide mode field mismatch. The attenuation is larger than 6.99 dB over the O-band, which shows a significant attenuation over the O-band. The maximum attenuation is 23.65 dB at 1279 nm. However, the attenuation is not as significant as the simulation. The reason for this is the deviation of the fabrication process and the loss from the waveguide deteriorating the performance of the VOA.

To measure the dynamic characteristics, a 1 kHz square wave signal was applied to the microheater, with two probes by a signal generator (SDG6032X-E, Siglent, China), and the output power was coupled into a high-speed photodetector. The driving voltage of the switch and the detected optical response were simultaneously observed on an oscilloscope (DS4024, RIGOL, China). Figure 4 shows the image of the oscilloscope with both the input driving signal (red) and the output optical signal (blue). The 10–90% rise and 90–10% fall time of the modulator are 184 μs and 180 μs, respectively.

## 4. Discussion

Table 1 shows a comparison of the VOAs reported in recent years, based on different platforms. Thanks to mature fabrication technology and edge polish technology, the VOA fabricated on a silicon platform shows ultra-low insertion loss. With air trenches introduced, the heat is insulated around cores, resulting in a lower power consumption (PC) and thermal crosstalk. However, the low thermo-optic coefficient of 1.19 × 10^–5^ K^−1^ contributes 155 mW of power consumption, which is about 15 times larger than a polymer-based VOA [12]. Moreover, the power consumption and crosstalk are decreased to 95 mW and 39 dB, since the air trench and etched-free waveguide also release the stress [13]. With suspended narrow ridge structure introduced, an ultralow power consumption of 20 mW is obtained based on a silica-based PLC platform [8]. However, this structure requires a somewhat-complicated fabrication process, including twice-dry etching and once-wet etching. Although the VOA [7] fabricated with silicon on insulator (SOI) shows fast response, the power consumption of 35mW is still a challenge for the large-scale integration with the compact footprint. In [14], a reflective MZI VOA consists of a 2 × 2 MMI coupler and a high reflection efficiency Bragg grating. The power consumption is 10.8 mW at 1580 nm. However, the VOA shows obvious wavelength dependence. The polymer-based devices show low power consumption, owing to a thermo-optic coefficient of 1.86 × 10^–4^ K^−1^. The polymer-based digital optical switch [9] shows a large ER of 45 dB, owing to a quartz substrate with grid patterns, which is also meaningful for our device. Moreover, air trench structure will decrease the power consumption shown in [20]. To improve response speed, a polymer/silica hybrid waveguide is introduced in [10]. Here, in our work, we adopt hybrid waveguides, achieving rise and fall times of lower than 200 μs. The large bandwidth of 100 nm makes the VOA able to cooperate with the CWDM system. The ER and PC could be further improved with grid pattern substrate and air trench.

## 5. Conclusions

In a summary, we demonstrate broadband polymer/silica hybrid waveguide thermo-optic VOA based on MZIs. We optimize MMIs, used in the VOA, to achieve wide bandwidth, low loss, and high fabrication tolerance performance. The VOA is fabricated through low-cost contact lithography and wet etching. At 1310 nm, an ER of 18.64 dB is achieved under 8.72 mW. The rise and fall time of the VOA are 184 μs and 180 μs, respectively. Over the O-band, the attenuation is larger than 6.99 dB. The VOA will be integrated with CWDM, for a power equalization function, monolithically.

## Figures and Tables

**Figure 1 micromachines-13-00511-f001:**
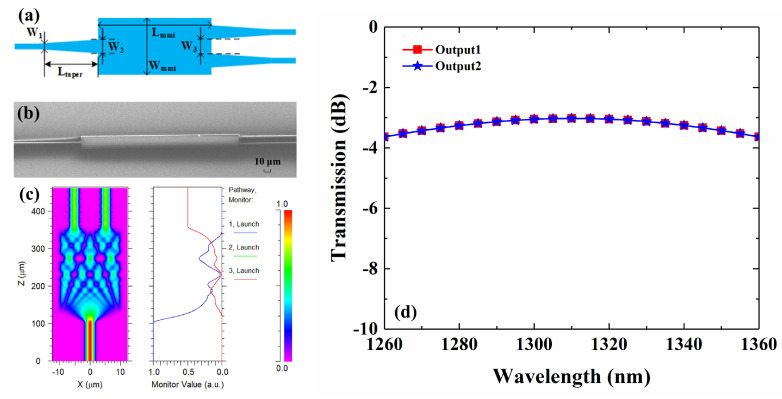
(**a**) The schematic of the optimized MMI; (**b**) SEM image of the fabricated MMI; (**c**) Calcu-lated field distribution along the MMI at 1310 nm; (**d**) Calculated spectra from the two output ports of the MMI.

**Figure 2 micromachines-13-00511-f002:**
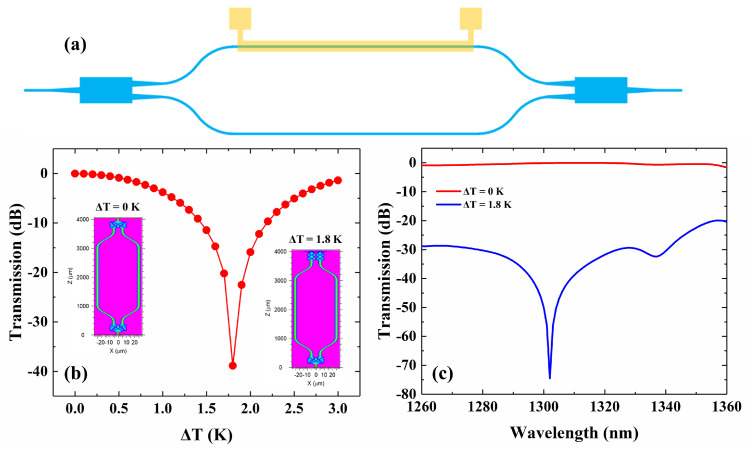
(**a**) The schematic of the VOA; (**b**) Calculated transmission versus modulation arm tem-perature change; (**c**) Calculated spectra of the VOA, while the temperature change of the modulation arm is 0 K and 1.8 K.

**Figure 3 micromachines-13-00511-f003:**
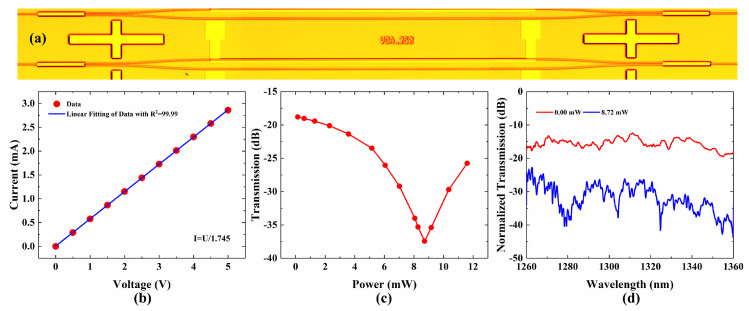
(**a**) Microscopic image of the proposed VOA; (**b**) Measured current-voltage (I-V) curves of the VOA; (**c**) Measured Transmission-Power (O-P) curves of the VOA; (**d**) Normalized transmission spectra of the VOA under 0.00 mW and 8.72 mW.

**Figure 4 micromachines-13-00511-f004:**
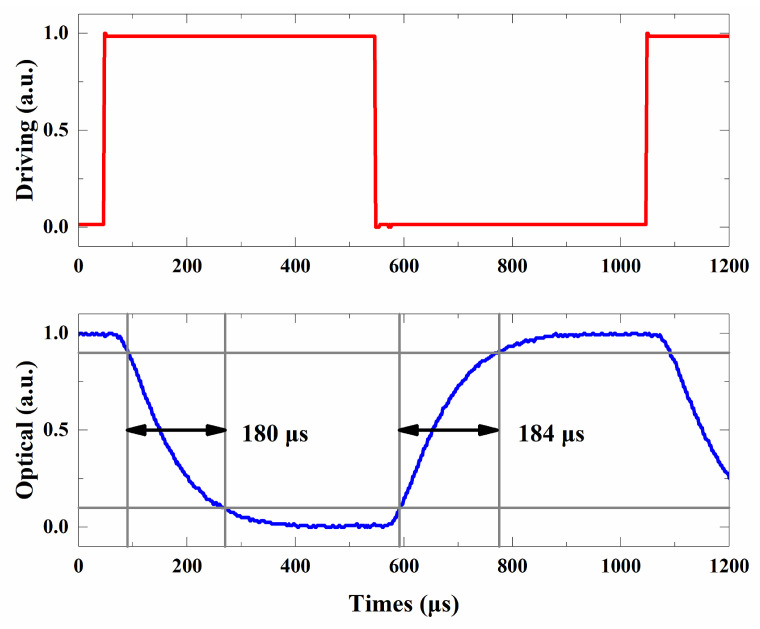
Time response of the broadband VOA.

**Table 1 micromachines-13-00511-t001:** Comparison of the VOA.

Ref.	Platform	IL (dB)	ER (dB)	PC (mW)	Work Range	RT (μs)	FT (μs)
[12]	Silica on silicon with air trench	0.7	30	155	1550 nm	N.A.	N.A.
[13]	Silica on silicon with	0.8	39	95	1550 nm	N.A.	N.A.
[8]	Silica on silicon with suspended narrow ridge structure	1.85	40	20	1539~1551 nm	5000	5000
[7]	Silicon on insulator	8.5	15	35	1550~1560 nm (10 nm)	2.8	7.1
[14]	Silicon on insulator	3.5	35.5	10.8	1580 nm	5	7
[9]	Polymer	1.35	45	50	1530~1560 nm (30 nm)	N.A.	N.A.
[20]	Polymer with air trench	9.3	40.2	2.8	1530~1610 nm (80 nm)	612	584
[10]	Polymer/silica with air trench	9.6	29.6	3.4	1550 nm	183.1	139.9
This work	Polymer/silica	12.43	18.64	8.72	1260~1360 nm (100 nm)	184	180

IL, insertion loss; ER, extinction ratio; PC, power consumption; RT, rise time; FT, fall time.

## Data Availability

Not applicable.

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
