# Peer review of "Polymer/Silica Hybrid Waveguide Thermo-Optic VOA Covering O-Band"

_micromachines, 2022, doi:10.3390/mi13040511_

Round 1
Reviewer 1 Report
Ref.comments to the paper “ Polymer/silica hybrid waveguide thermo-optic VOA covering O-band” written by the authors: Yue-Xin Yin, Meng-Ke Yao, Ying-Zhi Ding, Xin-Ru Xu, Yue Li, Yuan-Da Wu and Da-Ming Zhang.
It is known that currently, that in the field of innovative materials and devices based on them, the lithography method is often used, which allows reducing the number of technological operations and obtaining devices, including attenuators, with an improved design. From this point of view the current manuscript is interesting and useful.
For the first, it is remarked that the authors have made the literature search, analyzing 18 references. This indicates the knowledge of the problem, its useful application and finding ways to solve it. But, not so many papers written by the last 3 years are analyzed; only one paper on 2019 has been analyzed. Please extend the Introduction part including 5-7 papers published on 2017-2022 years in an area of your interest.
Design and Simulation section. Please indicate the type (number in catalog) of PMMA materials from Microchem Co. PMMA materials are so often used in the science and industry, but different types of them have the different parameters, for example free volume and refractive index. Illustration shown in figures 1 and 2 are good and permit to understand the text. Please tilt the Latin symbols in the text!
Fabrication and Characterizing section. Data from figure 3 - Microscopic image of the VOA proposed; Measured Current-Voltage (I-V) curves of
the VOA; Measured Transmission-Power (O-P) curves of the VOA; Normalized transmission spectra of the VOA under 0.00 mW and 8.72 mW – are coincided with our physical knowledge.
Discussion part. Information from Table 1 - Comparison of the Variable optical attenuator (VOA) – is interesting and useful not only for the science, but for the engineering people as well. I would like to ask the authors about the following aspect: Have you the data on the comparative parameters when the silica is covered with the carbon nanotubes with the refractive index close to 1,1?
Conclusion part should be dramatically extended!
So, the paper is interesting for the specific area for the researchers. The paper can be published after minor corrections according the questions shown before.
Reviewer 2 Report
It is good, clear and interesting presentation in the field of photonic devices, using simple hybrid technology for fabrication of thermo-optic VOA for O-band.
I have any critical comments.
